# Enhancing Niacinamide Skin Penetration via Other Skin Brightening Agents: A Molecular Dynamics Simulation Study

**DOI:** 10.3390/ijms26041555

**Published:** 2025-02-12

**Authors:** Kamolrat Somboon, Choon-Peng Chng, Changjin Huang, Shikhar Gupta

**Affiliations:** 1Bioinformatics Institute (BII), Agency for Science, Technology and Research (A*STAR), 30 Biopolis Street, #07-01 Matrix, Singapore 138671, Singapore; somboon_kamolrat@bii.a-star.edu.sg; 2School of Mechanical and Aerospace Engineering, Nanyang Technological University, Singapore 639798, Singapore; choonpeng.chng@ntu.edu.sg (C.-P.C.); cjhuang@ntu.edu.sg (C.H.); 3P&G Singapore Innovation Center (SgIC), 70 Biopolis Street, Singapore 138547, Singapore

**Keywords:** niacinamide, sucrose dilaurate, undecylenoyl phenylalanine, bisabolol, skin brightening, skin penetration

## Abstract

Niacinamide, a derivative of vitamin B3, has been shown to reduce skin pigmentation (i.e., acting as a brightening agent) and inflammatory responses such as dermatitis and acne vulgaris. However, niacinamide is a hydrophilic compound and poor partitioning to the lipid matrix in the uppermost layer of the skin (the stratum corneum or SC) limits its delivery to the skin. This necessitates the use of penetration enhancers to increase its bio-availability. In this study, we used computer simulations to investigate the skin penetration of niacinamide alone and in combination with other brightening agents that are also shown to be skin penetration enhancers, namely undecylenoyl phenylalanine (Sepiwhite^®^), bisabolol, or sucrose dilaurate. Molecular dynamics simulations were performed to reveal molecular interactions of these brightening agents with a lipid bilayer model that mimics the SC lipid matrix. We observed minimal penetration of niacinamide into the SC lipid bilayer when applied alone or in combination with any one of the three compounds. However, when all three compounds were combined, a notable increase in penetration was observed. We showed a 32% increase in the niacinamide diffusivity in the presence of three other brightening agents, which also work as penetration enhancers for niacinamide. These findings suggest that formulations containing multiple brightening agents, which work as penetration enhancers, may improve skin penetration of niacinamide and enhance the effectiveness of the treatment.

## 1. Introduction

Niacinamide (nicotinamide) is the heterocyclic aromatic amide form of water-soluble vitamin B3 (niacin). Vitamin B3 deficiency can lead to a disease known as Pellagra [1]. Niacinamide has been used for anti-hyperpigmentation/anti-aging applications [2,3], as an anti-oxidant [4], improving skin barrier function [5], and for alleviating some skin conditions such as acne vulgaris [6] and dermatitis [7]. Niacinamide reduces hyperpigmentation and improves skin brightness by inhibiting the transfer of melanosomes from melanocytes to keratinocytes as found in clinical studies [8,9]. By inhibiting poly (ADP-ribose) polymerase to down-regulate the expression of inflammatory mediators, niacinamide helps to reduce inflammatory responses such as dermatitis and acne vulgaris [6,7,10,11].

As niacinamide is highly soluble in water (AlogP of −0.32, Table 1), it is challenging to deliver it into the skin, where the uppermost layer of skin known as the stratum corneum (SC) consists of dead keratin cells embedded in a lipid matrix [12]. The hydrophobic lipid matrix represents the primary permeation pathway for any skincare active compound [13]. To overcome this barrier to skin permeation, various strategies have been explored and are either active or passive in nature. Active strategies to disrupt the SC barrier include electrical or mechanical means or the use of ultrasound or laser radiation [14]. Passive strategies include the use of simple solvent systems as skin penetration enhancers [15,16] and surfactants as emulsifying agents and as skin penetration enhancers [17]. Propylene glycol (PG) is a commonly used solvent, and it was shown that adding propylene glycol monolaurate to PG at a 1:1 ratio increased the skin flux and permeability coefficient of niacinamide by >400-fold and adding Miglyol 812N^®^ to the binary solvent further doubled the skin flux and permeability coefficient [15]. In another study, PG–linolenic and PG–oleic acid showed the highest cumulative permeation of niacinamide through human skin at 24 h, with a 100-fold enhancement compared to PG alone [16]. The fatty acids might have increased lipid fluidity prior to PG-promoted niacinamide penetration. Surfactants consist of a polar head and a non-polar tail which is either a straight or branched hydrocarbon or fluorocarbon chain with 8–18 carbon atoms [17]. Surfactants can be divided into natural, anionic, cationic, nonionic, and zwitterionic ones. A nonionic surfactant, Tween^®^80, in a dipropylene glycol solvent was found to enhance the skin permeability of niacinamide by about 13-fold compared to niacinamide in distilled water [18].

Molecular dynamics (MD) simulations are a useful tool to gain valuable molecular-level insights into drug permeation across the skin and how various penetration enhancers can facilitate the permeation. In MD simulations, the motions of various molecular species and their interactions are monitored in real time. MD simulations have been used to study the permeation of various small molecules through a skin lipid bilayer model composed of an equal molar ratio of Ceramide24 NS, C24 free fatty acid, and cholesterol [19]. The permeability of each molecule was computed and compared to experimental values, showing similar trends. The effect of chemical enhancers in modulating the permeability of various reference compounds has also been investigated by MD simulations [20,21]. Calculated permeability coefficients showed that the effect of various penetration enhancers depends on the compounds and their concentration [20]. The mechanism of the drug penetration enhancer propylene glycol (a simple solvent mentioned above) has also been studied via MD simulations, where a similar SC lipid bilayer composed of equal molar ratios of Ceramide24 NS, C24 free fatty acid, and cholesterol was used [22]. PG was found to localize to hydrophilic lipid headgroups and result in an increase in the area per lipid, with a slight increase in the lipid tail disorder in a concentration-dependent manner [22]. However, bilayer penetration by PG molecules was not observed even on microsecond simulation time scales despite evidence from experiments, likely due to energetic barriers of multiple kJ/mol that make spontaneous PG permeation a low probability event on the time scales of atomistic MD simulations.

Brightening molecules that are more amphiphilic (namely undecylenoyl phenylalanine or Sepiwhite^®^ (SPWD), bisabolol (BSB), and sucrose dilaurate (SDL) (see Table 1)) may enhance the permeation of niacinamide through the skin. SDL is a natural surfactant derived from sugars and fatty acids. As a sucrose ester, SDL is non-toxic and readily biodegradable. Sucrose laurate/dilaurate was shown to significantly reduce bilirubin levels in the skin, hence skin yellowness [23], and also enhance skin permeation of ibuprofen by over 2-fold [24]. Sucrose esters interact with SC lipids and fluidize the intercellular lipids thereby reducing the SC barrier function [25,26]. Sepiwhite^®^ is a commercial lipophilic derivative of phenylalanine that inhibits the activation of tyrosinase, an enzyme that converts L-tyrosine amino acid to melanin during melanogenesis [27]. Hyperpigmentation is a common skin problem with darkened patches or spots on the skin due to increased melanin production because of UV radiation, disease, or medications. Undecylenoyl phenylalanine was shown to be helpful in the treatment of melasma [28], a common skin pigmentary problem. A combined formulation of undecylenoyl phenylalanine and niacinamide was found to be significantly more effective than either the control vehicle or niacinamide alone in reducing hyperpigmentation over 8 weeks [29]. The same study also showed that undecylenoyl phenylalanine enhances the skin penetration of niacinamide, with over 50% of the applied niacinamide penetrating into the human skin over a 24 hr period [29]. Lastly, bisabolol is an unsaturated, optically active sesquiterpene alcohol from the essential oil of chamomile. It has anti-microbial and anti-skin inflammation properties [30,31]. Bisabolol is effective against skin inflammation by inhibiting the production of pro-inflammatory cytokines in macrophage cells and skin inflammation in mice [32] and was shown to significantly lighten the pigmented skin of Asian women in a clinical study [33]. Bisabolol was also demonstrated to enhance trans-epidermal drug penetration through the skin by possibly increasing lipid fluidity [34].

In this work, we aim to explore whether amphiphilic skin brightening agents, namely Sepiwhite^®^, bisabolol, and sucrose dilaurate, enhance the skin permeability of niacinamide by using computer simulations. These three brightening agents are hydrophobic as their 1-octanol/water partition coefficient values (AlogP) are greater than 4 (Table 1) which suggests a stronger preference for these three skin brightening agents in the lipid matrix than the hydrophilic niacinamide (Table 1). We performed MD simulations of niacinamide alone or in combination with the other compounds on a model SC lipid bilayer composed of equimolar ratios of Ceramide24 NS, free fatty acids, and cholesterol as used in previous simulation studies [19,22] but with three free fatty acids (C16:0, C18:0, and C20:0) in equimolar ratios to better represent the heterogeneity of free fatty acids in the SC. Previously, we reported the effect of palmitoylation (enhancing hydrophobicity) on the adsorption and diffusion of a short polar (hydrophilic) peptide across a model SC lipid bilayer using molecular dynamics simulations [35]. No significant penetration of niacinamide into the SC lipid bilayer was observed for niacinamide alone or together with either Sepiwhite^®^, bisabolol, and sucrose dilaurate on the simulation time scales. However, some degree of penetration into the SC lipid bilayer was observed for niacinamide in combination with all three compounds. The calculated niacinamide diffusivity across the SC lipid bilayer also showed higher average diffusivity in the presence of the enhancers. Our simulations thus suggest that these three compounds may act synergistically as penetration enhancers for niacinamide.

## 2. Results

### 2.1. Brightening Agents Adsorb onto the SC Lipid Bilayer

To evaluate how hydrophobic brightening agents interact with the SC lipid bilayer and influence niacinamide permeation, we carried out six separate MD simulations (Table 2). The simulations include a control solely featuring the SC lipid bilayer (None), a system with 5% niacinamide alone (NIA), and four systems combining 5% niacinamide with 2% bisabolol (NIA + BSB), 2% Sepiwhite^®^ (NIA + SPWD), 2% sucrose dilaurate (NIA + SDL), and a complex mixture of 5% niacinamide, 0.2% Sepiwhite^®^, 1% sucrose dilaurate, and 0.4% bisabolol (NIA + SPWD + SDL + BSB). We choose the concentration of different brightening agents based on one of the internal skincare formulations. The chemical structures of the SC lipid bilayer components, as well as the brightening agents, are illustrated in Figure 1. An atomistic model of the equilibrated SC lipid bilayer is also shown in Figure 1. Each simulation was run for 100 nanoseconds (ns) and performed in duplicate to ensure reproducibility.

To elucidate how brightening agents affect SC lipid bilayer integrity and packing, we monitored bilayer thickness, area per lipid, and hydrogen bond formation between the lipid headgroups of the SC lipid bilayer across different simulation systems (Table 2). Compared to the control system (None), the presence of brightening agents induced subtle alterations in the SC lipid bilayer structure. Specifically, the bilayer thickness showed fluctuations between 4.51 and 4.55 nm, indicative of minor perturbations in SC lipid bilayer packing. In contrast, the area per lipid remained largely unchanged across all systems, with values ranging from 0.98 ± 0.007 nm^2^ in NIA to 1.01 ± 0.021 nm^2^ in NIA + SPWD + SDL + BSB. This minimal variation suggests that brightening agents do not significantly disrupt the lateral packing of lipid tails, thereby maintaining the bilayer’s fundamental structural integrity. Additionally, analysis of hydrogen bond formation within the SC lipid bilayer revealed that the incorporation of brightening agents led to a slight increase in hydrogen bonds (Table 2). Specifically, the control system exhibited an average of 33.67 ± 0.21 hydrogen bonds, whereas systems containing niacinamide alone (NIA) and in combination with various brightening agents showed elevated hydrogen bond counts, ranging from 34.74 ± 1.32 to 37.48 ± 1.35 hydrogen bonds. These increases indicate a trend of enhanced hydrogen bond formation with the addition of brightening agents, suggesting that these agents facilitate additional molecular interactions within the SC lipid bilayer. Collectively, these findings demonstrate that brightening agents subtly modulate the structural and interactive properties of the SC lipid bilayer, contributing to an environment that supports niacinamide permeation without compromising SC lipid bilayer integrity.

In our simulations, visual inspections revealed that niacinamide did not fully traverse the SC bilayer, whereas other brightening agents partially embedded themselves within the bilayer (Figure 2). This partial embedding suggests the formation of transient interactions with lipid headgroups that can modify SC lipid bilayer properties. To quantify how deeply niacinamide could infiltrate the SC lipid bilayer over time without and with the other brightening agents, we monitored the minimum distance between niacinamide atoms and the center of the bilayer over time for each of the systems (Figure 3a). The shaded regions in the distance profiles represent the standard deviation at each time point, illustrating fluctuations in penetration depth. Notably, in systems containing multiple brightening agents (i.e., the mixture of niacinamide, bisabolol, Sepiwhite^®^, and sucrose dilaurate), niacinamide molecules showed the largest penetration (Figure 3a), with some of the other agent molecules penetrated more extensively into the bilayer core (Figure 3b). Niacinamide was observed to accompany SDL as it moved further into the bilayer interior at around 70 ns (Figure 3b), suggesting a cooperative effect wherein SDL alters local lipid packing, thereby facilitating niacinamide’s entry. Although niacinamide did not completely cross the bilayer within 100 ns, its capacity to reach deeper bilayer regions in the presence of brightening agents underscores its potential role in modulating permeability pathways.

### 2.2. Brightening Agents Absorbed onto SC Lipid Bilayer Enhance Diffusivity of Niacinamide Across the Bilayer

In our MD simulations, we investigated the influence of brightening agents on niacinamide diffusivity across the SC lipid bilayer under two primary conditions: a control system without any enhancers and a system containing these penetration-promoting agents. As depicted in Figure 4, the analysis focused on the central hydrophobic region of the bilayer from −2.3 nm to 2.3 nm (highlighted by the gray overlay), which represents the primary barrier region for molecular transport [19]. By averaging the values of the diffusivity profile D(z) over this region, we have derived an average diffusivity value that captures the overall mobility of niacinamide within the bilayer.

The control system without enhancers exhibited an average diffusivity of 2.41 × 10^−6^ cm^2^/s (Figure 4, left). In contrast, the enhancer-containing system showed a notable increase to 3.18 × 10^−6^ cm^2^/s (Figure 4, right), corresponding to a 32% enhancement. While this difference might seem modest in absolute terms, it underscores how small perturbations—such as slight disruptions in lipid packing or localized boosts in SC lipid bilayer fluidity—can facilitate both the lateral and transverse motion of niacinamide.

## 3. Discussion

Niacinamide acts as a skin brightening agent for anti-aging/anti-hyperpigmentation applications and is also helpful in reducing skin inflammatory responses such as dermatitis and acne vulgaris. Being a hydrophilic molecule, niacinamide does not partition easily into the hydrophobic lipid matrix of the skin stratum corneum (uppermost layer of the skin). Hence, skin-penetration-enhancing strategies such as the use of skin-penetration-enhancing molecules are required for the effective delivery of niacinamide into the skin.

We have used computer simulations to explore how niacinamide alone and in the presence of other brightening agents such as Sepiwhite^®^, bisabolol, and sucrose dilaurate may partition into a model of the skin SC lipid matrix. Brightening agent molecules were placed in the water phase above a lipid bilayer representing a unit of the SC lipid matrix in the stratum corneum. We found that niacinamide alone is not able to partition into the skin SC lipid bilayer and mostly stays in the water phase during the simulation. Niacinamide molecules were also observed to interact with the hydrophilic headgroup of SC lipids but could not penetrate into the hydrophobic lipid bilayer interior. In the presence of any one of the other brightening agents, there was also only minimal penetration of niacinamide into the lipid interior. However, in the presence of all three other brightening agents (SPWD, BSB, and SDL), a notable increase in lipid bilayer penetration of niacinamide was observed. Being amphiphilic in nature, SPWD, BSB, and SDL all have both hydrophilic headgroup and hydrophobic tail moieties which facilitate their partitioning into the SC lipid bilayer. These molecules were able to penetrate the SC lipid bilayer within the 100 ns time scale of our simulations, with SPWD and SDL inserting their hydrophobic chains into the SC lipid bilayer whereas the smaller BSB molecules were able to completely insert into the hydrophobic interior of the lipid bilayer as BSB has only one polar hydroxyl group (-OH). SDL has been suggested to interact with SC lipids and increase their fluidity thereby compromising the SC barrier function [25,26], with BSB also acting to increase lipid fluidity as it enhances drug penetration into the skin [34]. However, our simulations suggest that only when acting together do these three brightening agents significantly enhance the penetration of niacinamide that can be observed on our 100 ns time scale. Although niacinamide did not completely cross the SC lipid bilayer within 100 ns, its ability to reach deeper bilayer regions in the presence of brightening agents underscores the potential role that these agents collectively play in modulating skin permeability pathways.

From our simulations, we also computed the average diffusivity of niacinamide across the SC lipid bilayer interior using enhanced sampling techniques. We found that the average diffusivity increased by about 32% in the presence of the other three brightening agents which potentially acted as penetration enhancers for niacinamide. Though seemingly modest, this finding underscores the importance of small perturbations to lipid packing or localized increases in lipid fluidity via the insertion of amphiphilic molecules that can facilitate the motion of niacinamide into the lipid bilayer. These enhancements to niacinamide bilayer penetration are likely driven by transient interactions between the brightening agents and various regions of the bilayer, including both lipid hydrophilic headgroups and hydrophobic tail domains. By slightly altering the bilayer’s microstructure (1–2 percent changes to bilayer thickness and area per lipid were observed in our simulations), these agents might reduce the SC lipid bilayer’s resistance to solute diffusion, thereby supporting deeper penetration. Additionally, the hydrogen bonding propensity of the brightening agents (via carbonyl and hydroxyl groups) may help form more “fluid-like” or less rigid zones within the SC lipid bilayer, promoting niacinamide’s passage through otherwise tightly packed lipid regions.

In summary, incorporating the other brightening agents may act to enhance the bio-availability of each other and of niacinamide in the skin. Having multiple brightening agents in the same formulation further facilitates skin brightening via various biological pathways.

## 4. Materials and Methods

### 4.1. Generation and Equilibration of SC Lipid Bilayer Model

The SC lipid bilayer model employed in this study is similar to that used in our previous work where details can be found [35]. Briefly, our model of SC lipid bilayer was composed of 100 molecules of ceramide Cer24 NS (C24:0 fatty acid tail and C18:1 sphingosine tail), 100 molecules of cholesterol, and 100 molecules of protonated FFA (equal numbers of palmitic acid (C16:0), stearic acid (C18:0) and arachidic acid (C20:0)). The bilayer is generated by using PackMol version 20.14.3 software [36]. The CHARMM-GUI webserver (https://www.charmm-gui.org (accessed on 10 February 2025)) was used to generate model parameters based on the CHARMM36 additive force field as well as the best practices MD protocol [37,38]. The SC lipid bilayer model was then subjected to energy minimization, short equilibration MD runs with restraints on the lipid headgroup atoms progressively decreasing to zero, and finally, production simulation for 100 ns to generate an equilibrated SC lipid bilayer model. GROMACS MD simulation package version 2020.3 was used for the simulations [39,40,41]. System temperature was maintained at 310 K and system pressure was maintained at 1 bar. Electrostatic interactions were computed using the Particle Mesh Ewald method with a cut-off distance of 1.2 nm. Van der Waals (vdW) interactions were computed using a cut-off method with forces smoothly switched to zero between 1.0 and 1.2 nm.

### 4.2. Calculation of Biophysical Properties of the SC Bilayer

Biophysical properties of the equilibrated SC lipid bilayer were computed based on the simulation data obtained over 100 ns of the production run. The bilayer thickness was calculated based on the difference in the average Z-locations of Cer24 NS and arachidic acid headgroup atoms between the top and bottom monolayers as performed in our previous work [35] using Visual Molecular Dynamics (VMD) version 1.9.4a53 and Python scripts [42].

The area/lipid (APL) was computed using the FATSlim (Fast and Accurate Toolkit for lipid bilayer Simulations) version 0.2.2 program [43]. The APL was calculated as the average area of these Voronoi polygons over the equilibrated simulation frames. For this study, frames from the final 20 ns of the simulation (80–100 ns) were used to ensure analysis of equilibrated properties. The resulting APL values were averaged across all lipid molecules in the bilayer.

In addition to the bilayer thickness and APL, hydrogen bond interactions within the SC lipid bilayer were analyzed to understand the molecular interactions influencing lipid bilayer stability and permeability. Utilizing VMD’s built-in hydrogen bond analysis tool, hydrogen bonds formed between lipid molecules and brightening agents were identified based on a geometric cut-off of 3.0 Å for donor–acceptor distances. The hydrogen bond data generated were subsequently processed using an in-house Python script to calculate the total number of hydrogen bonds formed during each simulation. To ensure that the analysis reflected equilibrated interactions, only hydrogen bonds occurring within the final 20 ns (80–100 ns) of each simulation were averaged.

To evaluate niacinamide penetration into the SC lipid bilayer, we employed the GROMACS version 2022 tool *gmx mindist* to calculate the minimum distance between niacinamide molecules and the center of the SC bilayer throughout the simulation. This analysis was conducted for each replicate simulation, and the resulting minimum distance data from both replicates were averaged using a custom Python script. Specifically, the minimum distance between the niacinamide atoms and the midpoint of the bilayer was monitored over the 100 ns simulation period to assess the depth of penetration.

### 4.3. Generation of Brightening Agent Models and Simulation of Brightening Agent–SC Lipid Bilayer Interactions

The three-dimensional model of each compound (niacinamide, bisabolol, Sepiwhite^®^, and sucrose dilaurate) was generated using Accelrys Discovery Studio version 2024 and manually edited to match the required format for input into the CHARMM-GUI webserver which automatically generated the all-atom CHARMM36 force-field parameters for the compounds [37,38,44]. MD simulation of the compounds in aqueous solution (TIP3P water models) was first carried out to assess the stability of each model. After energy minimization to relax any possible steric clashes and 125 ps of simulation with compound heavy atoms restrained, unrestrained simulation was then carried out with a 2 fs simulation time step for 10 ns under a temperature of 310 K and pressure of 1 bar following the same protocol using the same associated parameters as for the SC lipid bilayer. Electrostatic and vdW interactions were also calculated using the same methods as for the SC lipid bilayer. For each system containing either niacinamide only or in the presence of the other compounds (see Table 2), the molecules were initially placed atop the SC lipid bilayer and solvated with TIP3P water molecules. For instance, the system containing all four compounds consists of fifty-four niacinamide, two bisabolol, one Sepiwhite^®^, and two sucrose dilaurate molecules. Energy minimization followed by short equilibration runs with positional restraints on compound heavy atoms and lipid headgroup atoms were carried out before unrestrained (production) runs were performed for 100 ns at 310 K and 1 bar for simulations of just the SC lipid bilayer. For post-processing, VMD version 1.9.4a53 software was used for visualization of the simulation trajectories, and GROMACS tools were used to calculate the temporal evolution of the minimum distance between the compound and midplane of the bilayer.

### 4.4. Calculation of Position-Dependent Diffusivity of Niacinamide Across the SC Lipid Bilayer

For the calculation of position-dependent diffusivity, a niacinamide molecule that was bound on the SC lipid bilayer was selected from the final simulation configuration and pulled both towards the bilayer center as well as away from the surface to generate configurations at various positions along the bilayer normal. Next, the niacinamide was then restrained at each of these positions with a force constant of 1000–3000 kJ/mol/nm^2^ and simulated for 40 ns (umbrella sampling method) similar to our previous work with skin targeting polar peptides [35]. Constraint forces were stored at every 0.1 ps during the last 10 ns of each sampling simulation and then used to calculate the position-dependent diffusivity in terms of a time correlation function involving constraint forces [19]:(1)$$D\left(z\right)=\frac{{\left(RT\right)}^{2}}{\int_{0}^{\infty }\left\langle ∆F\left(z,t\right)∆F\left(z,0\right)\right\rangle dt}$$
where R is the gas constant, T is the temperature in Kelvin, and ∆F(z,t) = F(z,t) − 〈F(z,t)〉 is the instantaneous deviation of the constraint force from the average value over time 〈F(z,t)〉. The time correlation function in the denominator of D(z) was written as a time integral over different starting times t′:(2)$$\left\langle ∆F\left(z,t\right)∆F\left(z,0\right)\right\rangle =\underset{T\to \infty }{\lim}\frac{1}{T}\int_{0}^{T}∆F\left(z,t^{‹}\right)∆F\left(z,t^{‹}+t\right)dt^{‹}$$

The algorithm to compute D(z) was implemented in Python using the NumPy library.

## 5. Conclusions

The brightening efficacy of brightening agents not only depends on their bio-activity but also on the bio-availability of all these brightening actives in the skin. We showed, through our simulations, that niacinamide, being a hydrophilic skin brightening agent, only interacts with the hydrophilic headgroup of SC lipids but could not penetrate the SC lipid bilayer interior, spending most of its time in water. The other three brightening agents (SPWD, BSB, and SDL) partially embedded themselves within the SC lipid bilayer during the simulation. Our simulation suggests that the penetration of niacinamide is significantly enhanced only when these three brightening agents act together. We showed a 32% increase in niacinamide average diffusivity in the presence of all three brightening agents which potentially acted as penetration enhancers and helped niacinamide to partition into the skin SC lipid matrix and become more bio-available in the skin for better efficacy. These brightening agents provide brightening efficacy via different biological pathways and incorporating these brightening agents together may act to enhance the bio-availability of each other and of niacinamide in the skin. Having multiple brightening agents in the same formulation further facilitates better skin brightening efficacy.

Our simulation study provides the importance of small perturbations to lipid packing or localized increases in lipid fluidity via the insertion of amphiphilic skin brightening molecules, which modulate skin permeability pathways and facilitate the partitioning of niacinamide into the lipid bilayer.

Collectively, these findings demonstrate that amphiphilic brightening agents subtly modulate the structural and interactive properties of the SC lipid bilayer, contributing to an environment that supports niacinamide permeation without compromising membrane integrity, which is very crucial to enhancing brightening efficacy without causing any adverse and harmful impacts on the skin.

## Figures and Tables

**Figure 1 ijms-26-01555-f001:**
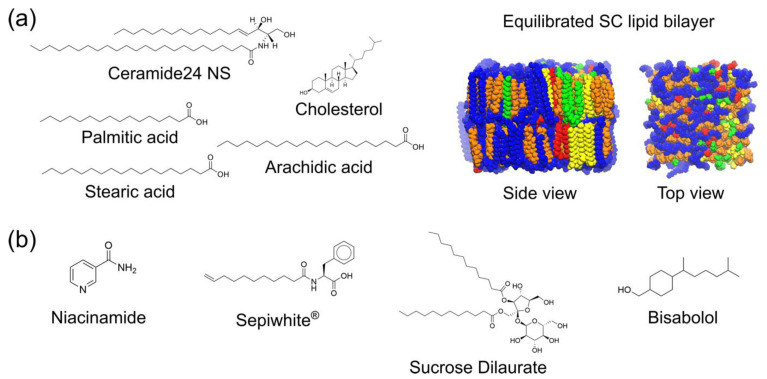
An all-atom model of the stratum corneum (SC) lipid bilayer interacting with brightening agents as penetration enhancers. (**a**) Chemical structures of the lipids present in our SC lipid bilayer model (left) and simulation snapshots showing the equilibrated SC lipid bilayer model as side and top views (middle and right). The number of Ceramide24 NS, cholesterol, and free fatty acid (FFA) molecules are present in an equimolar ratio. The three FFAs are also present in an equimolar ratio. Ceramides and cholesterol are colored blue and orange, respectively, while FFAs are colored as follows: palmitic acid in yellow, stearic acid in red, and arachidic acid in green. Water layers above and below the bilayer are omitted for clarity. (**b**) Chemical structures of brightening agents in this study.

**Figure 2 ijms-26-01555-f002:**
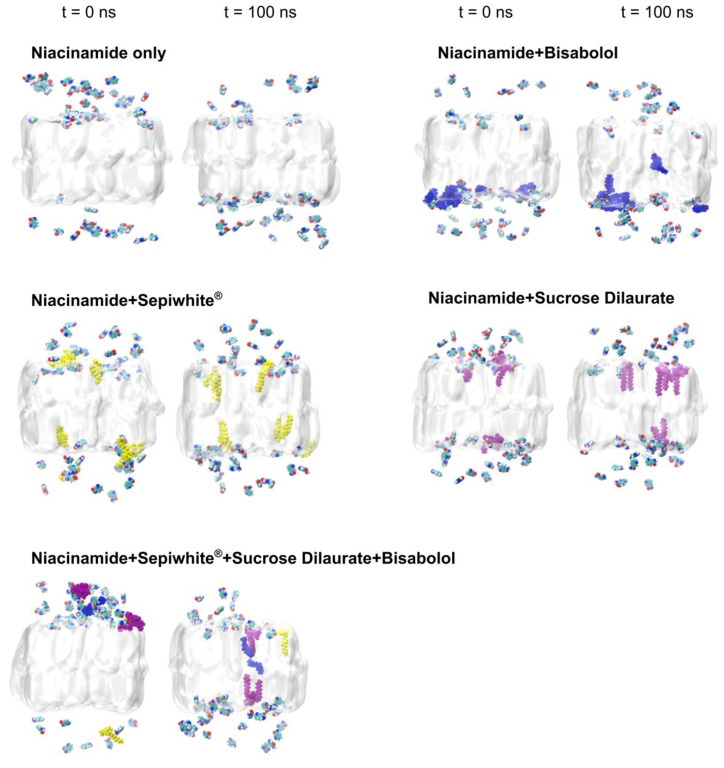
MD simulations of the interaction of brightening agents with the SC lipid bilayer. Simulation snapshots showing the initial (start of unrestrained production run, simulation time *t* at 0 ns) and final configurations of the brightening agents relative to the SC bilayer surface. Niacinamide is color-coded according to atom type, whereas bisabolol, Sepiwhite^®^, and sucrose dilaurate are represented in blue, yellow, and purple, respectively. The SC bilayer is shown in a transparent style for enhanced clarity.

**Figure 3 ijms-26-01555-f003:**
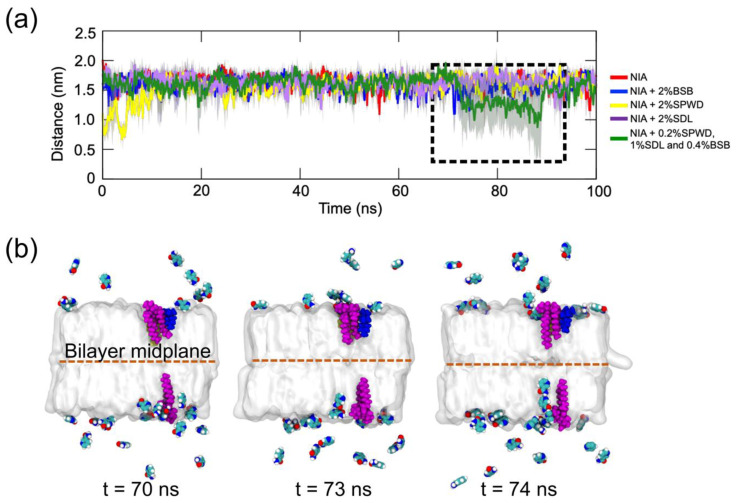
Penetration of the niacinamide molecules into the SC lipid bilayer in the presence of other brightening agents. (**a**) The temporal evolution of the minimum distance between niacinamide atoms and the middle of the SC lipid bilayer showing the depth of penetration of niacinamides in the different systems in Table 2. Shaded areas represent the standard deviation at each time point. (**b**) Selected simulation snapshots of the system with all three enhancers within a time window (dashed box in (a)) showing niacinamide molecules penetrating into the bilayer in proximity to SDL molecules (in purple, as in Figure 2) that were absorbed into the bilayer.

**Figure 4 ijms-26-01555-f004:**
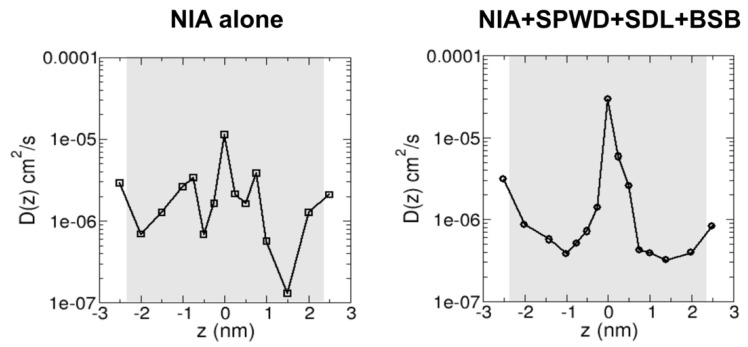
Diffusivity profiles of niacinamide across SC lipid bilayer in the absence or presence of other brightening agents as penetration enhancers. The average diffusivity values taken from −2.3 nm to 2.3 nm (within the gray overlay) of the profile without and with enhancers are 2.41 × 10^−6^ cm^2^/s and 3.18 × 10^−6^ cm^2^/s, respectively, with the latter value being 32% higher relative to the former.

**Table 1 ijms-26-01555-t001:** Chemical properties of brightening agents considered in this study. ALogP is an estimation of log P (the logarithm of 1-octanol/water partition coefficient). Molar volume is the ratio of molar mass to mass density.

Brightening Agent	ALogP	Molar Volume (m^3^/mol)
Niacinamide	−0.32	322
Sepiwhite^®^	5.19	852
Sucrose Dilaurate	5.99	1607
Bisabolol	4.31	593

**Table 2 ijms-26-01555-t002:** Biophysical properties of the SC lipid bilayer in the absence and presence of brightening agents as penetration enhancers. Niacinamide concentration is 5% by weight of the SC lipid bilayer. Average ± standard errors are reported based on two replicate simulations for each system.

Brightening Agent(s) Added	Bilayer Thickness (nm)	Area/Lipid (nm^2^)	# of Lipid Headgroup H-Bonds
None	4.53 ± 0.02	0.99 ± 0.004	33.67 ± 0.21
Niacinamide (NIA)	4.55 ± 0.00	0.98 ± 0.007	35.84 ± 1.18
NIA + 2% BSB	4.54 ± 0.00	1.01 ± 0.000	37.34 ± 0.94
NIA + 2% SPWD	4.51 ± 0.01	1.01 ± 0.000	34.74 ± 1.32
NIA + 2% SDL	4.54 ± 0.01	0.99 ± 0.002	36.72 ± 0.40
NIA + 0.2% SPWD, 1% SDL, and 0.4% BSB	4.51 ± 0.01	1.01 ± 0.021	37.48 ± 1.35

#: number of.

## Data Availability

Data may be made available upon reasonable written request to the corresponding author.

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
