# Peer review of "Enhancing Niacinamide Skin Penetration via Other Skin Brightening Agents: A Molecular Dynamics Simulation Study"

_ijms, 2025, doi:10.3390/ijms26041555_

Round 1

Reviewer 1 Report

Comments and Suggestions for Authors

In this manuscript, computer simulations were used to explore how niacinamide may partition into a model of the skin SC lipid matrix. The theme is sound and significance of the work is good. However, some issues should be addressed before publication in IJMS:

1. Lines 135-138: The authors should clarify the basis on which they selected these concentrations of brightening agents for performing the simulation.

2. Sepiwhite® should be defined the first time it appears in the text (Abstract) for clarity and to ensure readers are informed.

3. Line 151: There appears to be a typo error—"between the" is repeated and should be corrected

4. Lines 179-180: “…SC lipid bilayer over time without and without the other brightening agents”. I suppose you were thinking “with and without”?

5. Caption for Figure 3: The caption should be reviewed as “(b) Zoomed-in area in (a) for system…” is a bit unclear. The phrase “in (a)” should probably be omitted for clarity.

Author Response

Comment 1: Lines 135-138: The authors should clarify the basis on which they selected these concentrations of brightening agents for performing the simulation.

Response 1: Thanks for the comment; we have added the rational in the revised manuscript Line 147-148 about the selected concentration of brightening agents for performing the simulation:

“We choose the concentration of different brightening agents based on one of the internal skin care formulation.”

Comment 2: Sepiwhite® should be defined the first time it appears in the text (Abstract) for clarity and to ensure readers are informed.

Response 2: We have defined in the Sepiwhite (undecylenoyl phenylalanine) in the Abstract on Line 18:

“… namely undecylenoyl phenylalanine (Sepiwhite®), bisabolol or sucrose dilaurate.”

Comment 3: Line 151: There appears to be a typo error—"between the" is repeated and should be corrected

Response 3: Thanks for spotting the error. We have removed the repeated—"between the” from the manuscript Line 162.

Comment 4: Lines 179-180: “…SC lipid bilayer over time without and without the other brightening agents”. I suppose you were thinking “with and without”?

Response 4: The reviewer is correct. We have corrected it at Line 192.

Comment 5: Caption for Figure 3: The caption should be reviewed as “(b) Zoomed-in area in (a) for system…” is a bit unclear. The phrase “in (a)” should probably be omitted for clarity.

Response 5: We have modified the caption for Figure 3 (b) as follows:

“(b) Selected simulation snapshots of the system with all three enhancers within time window (dashed box in (a)) showing niacinamide molecules penetrating into the bilayer in proximity to SDL molecules (in purple) that were absorbed into the bilayer.”

Reviewer 2 Report

Comments and Suggestions for Authors

General comments

The manuscript presents experimental data of molecular dynamics simulation investigating the skin penetration by niacinamide, a known molecular with skin lightening properties, alone or in combination with other 3 compounds with similar potential: Sepiwhite (undecylenoyl phenylalanine), bisabolol and sucrose dilaurate. The authors conclude that niacinamide’s penetration alone was minimal in their experiments, whereas when combined with the other three components which may act as penetration enhancers, there was a 32% increase in the niacinamide’s penetration.

Specific comments

-          The chemical identity of Sepiwhite is mentioned on page 2. I understand that this is a commercially available compound sold under this brand name. However, it might be useful to include the chemical name in the abstract and then again when first mentioned in the Introduction as not all readers may be familiar with the brand name.

-          There is no clear justification provided as of why the other 3 compounds besides niacinamide were selected. Niacinamide itself is a well-known, published and formulated with compound and claimed to be very efficient in reducing skin pigmentation. The others are also reported to have such activities. I would like to ask the authors to share how they selected the compounds: did they scan the products on the market containing niacinamide and narrowed down to these 3 enhancers which also have melanin reducing activities or was the selection based on the most frequently used compounds in combination with niaciamide? My question has also a practical angle: in real life, in the lab, not all compounds can be formulated together. Some are soluble in unique solvents that are not acceptable in a cosmetic product, or when formulated together, the formula brakes apart due to incompatibilities or the formula is unstable in time.

-          Regarding the concentrations used for the simulation experiments: they are mentioned on page 3, section 2.10, lines 134-138. There is no mention as of why these particular concentrations were considered. Were they the ones mentioned in cosmetic products, for example? Were they selected based on the solubility of each compound? For example, niacinamide’s concentration in cosmetic products varies from 3% to 10%, so the authors selected a 5% which is somewhat in the middle of this range.

-          When mixed together, depending on the ratio used, solvents, pH, etc., compounds can interact with each other such that in a formula one might end with byproducts besides the parent compounds. I believe the simulations do not account for any such compounds resulting from the interaction of these 4 compounds when mixed together, but I wonder if the authors can speculate on the possibility of the compounds to interact with each other and if so, what resulting compounds may exist in the mix. Those resulting compounds may also contribute to the penetration and ultimately to the effects of the parent compounds.

-          It is very intriguing to me to learn from this manuscript that niacinamide does not partition easily into the hydrophobic lipid matrix of the stratum corneum because it is considered almost a “go to” molecule in the cosmetic industry; that said, I am not at all surprised since it is a hydrophilic compound. These experiments beg the question to the scientific community how does niacinamide actually impact pigmentation? From a penetration perspective and based on this paper, niacinamide cannot even reach the melanocytes in the skin which are located deep in the epidermis unless it is formulated with penetration enhancers. In this regard, even though not the immediate scope of this paper, I might encourage the authors to do a scan of the top 10 most frequently used skin lightening products containing niacinamide and identify if indeed enhancers are used, which would be the only explanation of its efficiency in reducing pigmentation. I am not asking for these data to be included in the paper since it would address existing products that should not be presented with the commercial names to avoid any conflicts on marketed products, but I consider this a worthwhile analysis for their own knowledge if they consider using their platform as a screening tool (more notes on this below).

-          I think the authors should make an effort to integrate more the key role of Sepiwhite, bisabolol and sucrose dilaurate for formulators to consider as enhancers besides the ones that might be regularly used already. Last but not least, these enhancers might act through a different mechanism of action when impacting the melanin production, which is desirable by cosmetic companies that are always looking for the best next compound.

-          From a practical perspective, do the authors consider their experiments as a “pre-screening” tool that cosmetic industry might use when considering new formulations from an efficacy perspective? I think the authors need to integrate their work in the field from a practical perspective, going beyond the academic exercise. For example, would this type of experiment be affordable financially before embarking on elaborated lab and clinical experiments that might not demonstrate in the end efficacious compounds reducing pigmentation that could have been anticipated by a simulation experiment?

-          What are the future plans for this strategy, again from an applicability perspective? For example, would the authors consider taking a product from the market and compare the penetration of that product containing niacinamide and some other components to the penetration of their combination used in this manuscript? For the paper to have a practical impact and attract interest from the scientific community, it needs to be integrated in such a way that it is considered a practical, useful tool guiding formulators. Otherwise, it would remain an elaborated experiment that has no or limited applicability.

Author Response

Comment 1:  The chemical identity of Sepiwhite is mentioned on page 2. I understand that this is a commercially available compound sold under this brand name. However, it might be useful to include the chemical name in the abstract and then again when first mentioned in the Introduction as not all readers may be familiar with the brand name.

Response 1: We thank the review for the suggestion. We have defined in the Sepiwhite (undecylenoyl phenylalanine) in the Abstract on Line 18:

“… namely undecylenoyl phenylalanine (Sepiwhite®), bisabolol or sucrose dilaurate.”

And also on Line 88 in the Introduction in the revised manuscript:

“Brightening molecules which are more amphiphilic (namely undecylenoyl phenylalanine or Sepiwhite® (SPWD),…”

Comment 2:  There is no clear justification provided as of why the other 3 compounds besides niacinamide were selected. Niacinamide itself is a well-known, published and formulated with compound and claimed to be very efficient in reducing skin pigmentation. The others are also reported to have such activities. I would like to ask the authors to share how they selected the compounds: did they scan the products on the market containing niacinamide and narrowed down to these 3 enhancers which also have melanin reducing activities or was the selection based on the most frequently used compounds in combination with niaciamide? My question has also a practical angle: in real life, in the lab, not all compounds can be formulated together. Some are soluble in unique solvents that are not acceptable in a cosmetic product, or when formulated together, the formula brakes apart due to incompatibilities or the formula is unstable in time.

Response 2: Thanks for the nice question! We selected the other 3 skin brightening actives Sepiwhite (undecylenoyl phenylalanine), Sucrose dilaurate, and bisabolol along with niacinamide based on the one of our internal skin care formulations. These actives are working via different pathways in a synergistic manner to enhance the overall brightening efficacy. Previously it was demonstrated that the combined formulation of Sepiwhite and niacinamide was found to be significantly more effective than niacinamide alone in reducing hyperpigmentation over 8 weeks [31]. The same study also showed that undecylenoyl phenylalanine enhances the skin penetration of niacinamide, with over 50% of the applied niacinamide penetrated into the human skin over a 24-hrs period [31]. So far, all 4 brightening ingredients have been formulated in the skin care products successfully and going to be in the market soon.

Comment 3: Regarding the concentrations used for the simulation experiments: they are mentioned on page 3, section 2.10, lines 134-138. There is no mention as of why these particular concentrations were considered. Were they the ones mentioned in cosmetic products, for example? Were they selected based on the solubility of each compound? For example, niacinamide’s concentration in cosmetic products varies from 3% to 10%, so the authors selected a 5% which is somewhat in the middle of this range.

Response 3: Thanks for the comment. We have added the rational in the revised manuscript Line 147-148 about the selected concentration of brightening agents for performing the simulations. Basically we choose the concentration of different brightening agents based on one of the internal skin care formulation.

Comment 4: When mixed together, depending on the ratio used, solvents, pH, etc., compounds can interact with each other such that in a formula one might end with byproducts besides the parent compounds. I believe the simulations do not account for any such compounds resulting from the interaction of these 4 compounds when mixed together, but I wonder if the authors can speculate on the possibility of the compounds to interact with each other and if so, what resulting compounds may exist in the mix. Those resulting compounds may also contribute to the penetration and ultimately to the effects of the parent compounds.

Response 4: Thanks for the comment. So far, all 4 brightening ingredients have been formulated in the skin care products successfully. This is the formulation craftmanship that we have chosen the ratios, solvents and pH such that no interactions among the brightening actives were observed. Also as you’ve mentioned, classical molecular dynamics simulations do not account for any such byproduct compounds resulting from the interaction of these 4 compounds when mixed together. The formulation is quite stable and do not have any stability issue as well as observation that all brightening actives are found to be very stable and in their original form even after the 6 months of stability testing as per standard cosmetics stability  testing conditions.

Comment 5: It is very intriguing to me to learn from this manuscript that niacinamide does not partition easily into the hydrophobic lipid matrix of the stratum corneum because it is considered almost a “go to” molecule in the cosmetic industry; that said, I am not at all surprised since it is a hydrophilic compound. These experiments beg the question to the scientific community how does niacinamide actually impact pigmentation? From a penetration perspective and based on this paper, niacinamide cannot even reach the melanocytes in the skin which are located deep in the epidermis unless it is formulated with penetration enhancers. In this regard, even though not the immediate scope of this paper, I might encourage the authors to do a scan of the top 10 most frequently used skin lightening products containing niacinamide and identify if indeed enhancers are used, which would be the only explanation of its efficiency in reducing pigmentation. I am not asking for these data to be included in the paper since it would address existing products that should not be presented with the commercial names to avoid any conflicts on marketed products, but I consider this a worthwhile analysis for their own knowledge if they consider using their platform as a screening tool (more notes on this below).

Response 5: Thanks for the suggestions and glad to know that our manuscript is providing the meaningful use of other skin brightening actives as penetration enhancer for niacinamide to reach up to the epidermis and provide better anti-pigmentation benefits to the consumers. Surely in future work we may scan the top 10 skin lightening products and identify the different penetration enhancers for niacinamide to enhance our understanding in this space.

Comment 6: I think the authors should make an effort to integrate more the key role of Sepiwhite, bisabolol and sucrose dilaurate for formulators to consider as enhancers besides the ones that might be regularly used already. Last but not least, these enhancers might act through a different mechanism of action when impacting the melanin production, which is desirable by cosmetic companies that are always looking for the best next compound.

Response 6: Absolutely correct! Sepiwhite, bisabolol and sucrose dilaurate work on different biological pathways to impact melanin production in the skin. But these can also act as niacinamide skin penetration enhancers as uncovered by our simulation work. We are educating our formulators to use these 3 compounds as skin penetration enhancers for niacinamide and other hydrophilic skin care actives without using the traditional penetration enhancers which might cause skin irritation to the consumers.

Comment 7: From a practical perspective, do the authors consider their experiments as a “pre-screening” tool that cosmetic industry might use when considering new formulations from an efficacy perspective? I think the authors need to integrate their work in the field from a practical perspective, going beyond the academic exercise. For example, would this type of experiment be affordable financially before embarking on elaborated lab and clinical experiments that might not demonstrate in the end efficacious compounds reducing pigmentation that could have been anticipated by a simulation experiment?

Response 7: Yes, we are indeed using simulations as pre-screening tools for creating new skin care formulations as this is enhancing our speed of innovation as well as providing new and non-obvious formulation insights.

Comment 8: What are the future plans for this strategy, again from an applicability perspective? For example, would the authors consider taking a product from the market and compare the penetration of that product containing niacinamide and some other components to the penetration of their combination used in this manuscript? For the paper to have a practical impact and attract interest from the scientific community, it needs to be integrated in such a way that it is considered a practical, useful tool guiding formulators. Otherwise, it would remain an elaborated experiment that has no or limited applicability.

Response 8: Thank you for the suggestion. Indeed we are planning to use this tool as screening criteria for creating any new formulations. Also, we are going to look at other in-market products containing niacinamide with other penetration enhancers and our combination in the manuscript to compare the impact on niacinamide skin penetration.